# Mitochondrial protein BNIP3 regulates Chikungunya virus replication in the early stages of infection

**Liliana Echavarria-Consuegra**[1¤a]*, **Nilima Dinesh Kumar**[1,2], **Marleen van der Laan**[1], **Mario Mauthe**[2], **Denise Van de Pol**[1], **Fulvio Reggiori**[2¤b], **Jolanda M. Smit**[1]*

**1** Department of Medical Microbiology and Infection Prevention, University Medical Center Groningen, University of Groningen, Groningen, The Netherlands, **2** Department of Biomedical Sciences of Cells & Systems, University Medical Center Groningen, University of Groningen, Groningen, The Netherlands

¤a Current address: Division of Virology, Department of Pathology, University of Cambridge, Addenbrooke's Hospital, Cambridge, United Kingdom
¤b Current address: Department of Biomedicine, Aarhus University, Aarhus, Denmark
* sle47@cam.ac.uk (LE-C); jolanda.smit@umcg.nl (JMS)

## Abstract

Chikungunya virus (CHIKV) is a human pathogen causing outbreaks of febrile illness for which vaccines and specific treatments remain unavailable. Autophagy-related (ATG) proteins and autophagy receptors are a set of host factors that participate in autophagy, but have also shown to function in other unrelated cellular pathways. Although autophagy is reported to both inhibit and enhance CHIKV replication, the specific role of individual ATG proteins remains largely unknown. Here, a siRNA screen was performed to evaluate the importance of the ATG proteome and autophagy receptors in controlling CHIKV infection. We observed that 7 out of 50 ATG proteins impact the replication of CHIKV. Among those, depletion of the mitochondrial protein and autophagy receptor BCL2 Interacting Protein 3 (BNIP3) increased CHIKV infection. Interestingly, BNIP3 controls CHIKV independently of autophagy and cell death. Detailed analysis of the CHIKV viral cycle revealed that BNIP3 interferes with the early stages of infection. Moreover, the antiviral role of BNIP3 was found conserved across two distinct CHIKV genotypes and the closely related Semliki Forest virus. Altogether, this study describes a novel and previously unknown function of the mitochondrial protein BNIP3 in the control of the early stages of the alphavirus viral cycle.

## Author summary

Chikungunya virus causes an acute febrile disease for which no vaccines and therapy is available. The virus is transmitted to humans via infected *A. aegypti* or *A. albopictus* mosquitoes who are mainly circulating in the (sub)tropical regions of the world. For the identification of antiviral targets it is important to understand the interactions between the virus and the host cell during infection. In this study we assessed the role of autophagy-related proteins in chikungunya virus infection. We found that the host cell factor mitochondrial protein and autophagy receptor BCL2 Interacting Protein 3 (BNIP3) controls

**Data Availability Statement:** All relevant data are within the manuscript and its Supporting Information files.

**Funding:** F.R. is supported by ZonMW TOP (91217002), Open Competition ENW-KLEIN (OCENW.KLEIN.118) and Marie Skłodowska Curie ETN (765912) grants. J.S., F.R. and N.D.K. were supported by a Skłodowska-Curie Cofund (713660). L.E. was supported by Erasmus Mundus EURICA mobility programme and Graduate School of Medical Sciences PhD scholarship (University Medical Center Groningen). M.M. and D.v.d. Pol were supported by the University Medical Center Groningen. The funders had no role in study design, data collection and analysis, decision to publish, or preparation of the manuscript.

**Competing interests:** The authors have declared that no competing interests exist.

the infectivity of chikungunya virus. Subsequent analysis revealed that BNIP3 interferes with chikungunya virus infection during the early stages of virus replication. The exact mechanism is not yet understood although we demonstrate that BNIP3 functions independent of its known roles in autophagy and cell death.

## 1. Introduction

Chikungunya virus (family *Togaviridae*) is a re-emerging alphavirus that is transmitted to humans by blood-feeding *Aedes* spp. mosquitoes and causes outbreaks of febrile illness in countries around Asia, Africa, Europe and the Americas [1–3]. Acute illness caused in humans by CHIKV is characterized by fever, headache, nausea, and severe joint and muscle pain. Although chikungunya fever resolves within 1–2 weeks after infection, the disease progresses to a chronic debilitating illness that affects the joints and can persist for months to years in about 35–52% of the infected people [4]. Current approaches to treat CHIKV-infected patients only involve unspecific therapies to relieve disease symptoms, since specific antivirals or licensed vaccines remain unavailable [5–7]. Thus, a better understanding of virus replication and disease pathogenesis might aid in the development of novel antiviral strategies.

CHIKV is an enveloped virus with a single-stranded positive-sense RNA genome. The RNA genome encodes four non-structural proteins (nsP1 to nsP4) and five structural proteins (the capsid protein, the accessory protein 6K/TF, the envelope proteins E1, E2 and E3), which are encoded by two open reading frames (ORF) [8,9]. Attachment of CHIKV E2 glycoprotein to cellular receptors triggers the internalization of the virus particle via clathrin-mediated endocytosis [10]. Upon fusion with the early endosome limiting membrane, the nucleocapsid is released into the cytoplasm and the first ORF is translated into non-structural proteins [10,11]. RNA replication occurs in bulb-like structures on the plasma membrane known as spherules, and it is driven by the non-structural proteins, which form the viral replicase complex [12]. The viral replicase synthesises a second ORF, also referred to as subgenomic RNA, from which the structural proteins are translated [13,14]. The E1/E2/E3 viral envelope proteins are translocated into the endoplasmic reticulum (ER) membrane and are processed during their transport to the plasma membrane to form the mature E1/E2 viral spike proteins. In parallel, progeny genomic RNA is packaged by the capsid protein to form the nucleocapsid [15]. Interaction of the newly formed nucleocapsid with the viral E1/E2 spike proteins at the plasma membrane drives the assembly, budding and release of the progeny virus particle [15].

As obligatory parasites, viruses depend on the host cell for replication and spread. Therefore, they have evolved mechanisms to exploit host cell pathways and proteins to promote their viral cycle. However, host factors orchestrate antiviral responses that counteract viral replication [16,17]. Autophagy is a lysosomal recycling pathway that involves the formation of double-membrane vesicles, known as autophagosomes, which sequester cytoplasmic cargoes targeted for degradation [18]. Specific cargoes, such as mitochondria or ER fragments, are selectively recognized by autophagosomes via the so-called autophagy receptors [19]. Previous research on the involvement of autophagy in CHIKV infection has revealed that BECLIN1 (BECN1) and ATG7, key players in the initiation and elongation of autophagosomes, promote CHIKV in HeLa and Hek293 cells, respectively [20,21]. Additionally, autophagy receptor nuclear dot 10 protein 52 (NDP52), plays a positive role in CHIKV infection, while the autophagy receptor sequestosome 1 (SQSTM1/p62) can suppress viral infectivity via autophagy [20,21]. These contrasting results for autophagy related (ATG) proteins that engage in the same pathway suggests that they could participate in cellular processes other than the

nucleation and maturation of autophagosomes. In fact, increasing evidence shows that many ATG proteins function in cellular process that are unrelated to autophagy [22,23]. For instance, we have previously reported that ATG13 and FIP200 play a role in picornavirus replication independently of their function in the formation of autophagosomes as a part of the ULK kinase complex [24].

Here, we used an image-based siRNA screen approach to comprehensively assess the role of 50 ATG proteins and autophagy receptors in CHIKV replication in human U2OS cells. Downregulation of 6 out of 50 ATG proteins was found to promote CHIKV replication suggesting that autophagy, as a cellular pathway, counteracts infection. The most pronounced effect was observed for BNIP3, a protein that mediates both the selective autophagic clearance of mitochondria, i.e., mitophagy [25] and cellular death programmes [26,27]. We found that downregulation of BNIP3 increases CHIKV infection, thereby leading to an enhanced production of progeny virus particles. This novel function of BNIP3 in controlling CHIKV infection is independent of autophagy/mitophagy and is not associated with CHIKV-induced cell death. We demonstrate that BNIP3 interferes with infection after virus membrane hemifusion but prior to abundant RNA replication. These antiviral properties of BNIP3 are conserved between circulating CHIKV strains of distinct genotypes and can be extended to the closely related alphavirus Semliki Forest virus (SFV).

## 2. Materials and methods

### Primary antibodies and reagents

The following primary antibodies were used in this study: anti-RFP (mouse monoclonal, 6G6, Chromotech), anti-BNIP3 (mouse monoclonal, ANa40, Santa Cruz), anti-BNIP3 (rabbit, monoclonal, EPR4034, Abcam), anti-LC3 (rabbit polyclonal, Novus biologicals), anti-ATG7 (rabbit polyclonal, 2631, Cell Signaling Technology), anti-Tom20 (rabbit polyclonal, FL-145, Santa Cruz), anti-GAPDH (mouse monoclonal, 6C5, Abcam), anti-tubulin (mouse monoclonal, Sigma), anti-vinculin (mouse monoclonal, hVIN1, Sigma-Aldrich), and anti-nsP2 (mouse monoclonal, ABM3F3.2E10, Abgenex). Rabbit polyclonal antibodies targeting CHIKV-E1, CHIKV-Capsid and the CHIKV-E2 stem proteins were kindly provided by G. Pijlman (Wageningen University, The Netherlands).

Menadione (M5625) solubilized in DMSO and deferiprone (DFP, 379409) solubilized in water, were purchased from Sigma-Aldrich (Merck), and used at a final concentration of 100 μM and 1 mM, respectively. Bafilomycin A$_1$ (BafA1, Bioaustralis) stock was prepared in DMSO and used at a working concentration of 100 nM. The mitochondrial uncoupler carbonyl cyanide m-chlorophenyl hydrazone (CCCP, Sigma) was used at a final concentration of 10 μM. Starvation-induced autophagy was triggered by incubation with Hanks' balanced salt solution (HBSS, 24020117, Thermo Fisher scientific).

### Cell lines

U2OS cells (ATCC HTB-96), *ATG7* knockout (ATG7KO) U2OS [28], HeLa cells (ATCC CRM-CCL2) and HuH-7 cells were maintained in DMEM, high glucose, GlutaMAX Supplement (Gibco). U2OS cells stably expressing GFP-WIPI2B were kindly provided by T. Proikas-Cezanne, University of Tübingen, Germany) [29] and were cultured in medium containing 600 μg/ml of G418 Sulfate (Gibco). MRC-5 cells (ATCC CCL-171) were maintained in DMEM/F12 (Gibco). A pCDNA3.1 encoding for COX8-EGFP-mCherry was generated by subcloning the COX8-EGFP-mCherry coding region from the pCLBW plasmid, a gift from D. Chan (Addgene plasmid #78520), and transfected into U2OS cells using Fugene HD (Promega Madison, WI) [30]. COX8-EGFP-mCherry-expressing U2OS cells were then selected with

800 μg/ml of G418 sulfate (Gibco). An expression vector encoding for BNIP3 fused to mCherry (pmCherry C1-BNIP3) was kindly provided by I. Novak (University of Split, Croatia). U2OS cells stably expressing BNIP3-mCherry were generated by transfection with Lipofectamine 3000 (Invitrogen) and selection with G418 sulfate (Gibco) as described above. Green monkey kidney Vero-WHO (ECACC 88020401) and Vero-E6 (ATCC CR-1586) cells were cultured in high glucose DMEM (Gibco), while BHK-21 (ATCC CCL-110) were cultured in RPMI 1640 medium (Gibco). All media were supplemented with 10% heat inactivated foetal bovine serum (Lonza) and 100 U/ml penicillin/streptomycin (Gibco). Cells were maintained at 37°C in humidified atmosphere containing 5% $CO_2$.

## Virus production and titration

Wild type (WT) CHIKV-La Reunion (LR, OPY1 strain) and 5'GFP-CHIKV-LR were produced from infectious cDNA clones generously provided by A. Merits (University of Tartu, Estonia) and from European Virus Archive goes global (EVAg) [31], respectively. CHIKV strain S27 was a kind gift from S. Guenther (Bernhard-Nocht-Institute for Tropical Medicine, Germany) and CHIKV strain 99659 was provided by M. Diamond (Washington University, St. Louis, USA). SFV was obtained from M. Kielian (Albert Einstein College School of Medicine, New York, USA). For viruses derived from an infectious clone, plasmid DNA was linearized with NotI (Invitrogen) and SP6 RNA polymerase (NEB) was used to synthesize RNA. The *in vitro* transcribed RNA was subsequently electroporated (Bio-Rad gene pulser Xcell machine; 850V, 25 μF, no resistance) into BHK-21 cells and virus-containing supernatants were collected at 48 h post-electroporation. To generate the working virus stocks, sub-confluent monolayers of Vero-E6 cells for the CHIKV-LR infectious clones or Vero-WHO cells for SFV and CHIKV-S27, were infected at a multiplicity of infection (MOI) 0.01. At 48 h post-infection (hpi), the cell supernatants were harvested and clarified from cell debris by low-speed centrifugation, snap frozen in liquid nitrogen and stored at -80°C. Infectious virus titres were determined by standard plaque assay on Vero-WHO cells for CHIKV or Vero-E6 cells for SFV. The number of CHIKV genome equivalent copies present in solution was determined by RT-qPCR, by amplification of the E2 gene, as previously described [32].

## siRNA based screen

For the primary siRNA screen, a customized ON-TARGETplus SMARTpool human siRNA library (Dharmacon, Horizon) targeting 50 ATG genes (S1 Table) was used to reverse transfect U2OS cells in a 96-well plate format. 2 pmol of siRNA were used for single gene knock-down, whereas for multiple gene knockdowns 3 pmol, 4.5 pmol, or 6 pmol were used for silencing of 2, 3 or 4 genes, respectively. Reverse transfection was performed using 0.1 μl of Lipofectamine RNAiMax transfection reagent (Invitrogen) per well, according to the manufacturer's protocol. After 20 min, 3,000 U2OS cells were seeded per well in complete culture medium to a final volume of 100 μl. At 48 h post-transfection (hpt), cells were infected with 5'GFP-CHIKV-LR at MOI 10. At 17 hpi, cells were fixed with 4% paraformaldehyde (PFA) and nuclei stained with Hoechst33342 (Sigma-Aldrich).

Images were acquired in a Cellomics ArrayScan VTI HCS Reader (Thermo Fisher scientific) at the Cell Screening Core of the University Medical Centre Utrecht, The Netherlands. For image acquisition, the Hoechst, FITC filters and 10x lens were used. Autofocus and cell number were determined in the Hoechst channel, and the FITC filter was used to detect GFP. An equal fixed exposure time was automatically set for all the samples. For image analysis, the Cellomics SpotDetector V3 algorithm was used to determine GFP expression in approximately 1,500 cells per well. Two parameters were measured and calculated: 1) percentage of GFP

positive cells, and 2) GFP intensity per well. Median Z-score values were used to standardize the measurements of each parameter from 4 independent experiments. Z-score values below −1.96 and above 1.96 were considered as statistically significant, with $p < 0.05$. To measure the extent of the variation from the control, the ratios of each parameter versus the non-targeting siRNA (siScramble)-transfected cells were calculated for each knockdown.

### Knockdown efficiency by siBNIP3 in mCherry-BNIP3-expressing cells

U2OS cells transiently expressing either pmCherry C1-BNIP3 or an empty pcDNA-mCherry vector were transfected with the siScramble or the siRNA targeting BNIP3 for 16 h. A shorter time-point after siRNA transfection was chosen for analysis as extensive cell death at later time-points was observed in cells when BNIP3 was overexpressed but not silenced. Cells were harvested and processed for flow cytometry and western blot as described below.

### siRNA-mediated silencing and infection

Individual or pooled siRNAs targeting BNIP3 (ID: LU-004636-00), ATG7 (ID: J-020112-08-0002) or NIX (ID: L-011815-00-0005), or non-targeting siRNA (siScramble, ID: D-001810-10) were obtained from Dharmacon (Horizon). Reverse and forward transfection of U2OS, ATG7 KO U2OS, MRC-5, HeLa, and HuH-7 cells was performed using Lipofectamine RNAiMAX (Invitrogen) with a final siRNA concentration of 20 nM according to manufacturer's protocol in either Nunc Lab-Tek II Chambered Coverglass, 24 well or 6 well plates. At 48 hpt, cells were infected with CHIKV or SFV at the indicated MOI.

### Flow cytometry

Cells infected with CHIKV were harvested at the indicated time points by detachment with 0.5% trypsin-EDTA (Gibco). Thereafter, the cells were fixed with 4% PFA and suspended in FACS buffer (PBS, 5% FBS, 1% EDTA). To detect infected cells, GFP or CHIKV E2 surface protein expression was assessed by flow cytometry. In the latter case, secondary staining was performed using the Alexa 647-conjugated goat anti-rabbit antibody (Life Technologies).

Cell death assays were carried out using a staining protocol based on the eBioscience Annexin-V apoptosis detection kit PE and Fixable Viability Dye eFluor 780 (FVD; Invitrogen), following manufacturer's recommendations.

In all experiments, 10,000 to 30,000 events were acquired per condition in a BD FACSVerse or a BD-LSR-II instrument (BD Biosciences), and analysed using the Kaluza Analysis software, Version 2.1 (Beckman Coulter).

### Mitochondrial mass measurement

To evaluate mitochondrial mass by flow cytometry, U2OS cells were incubated for 25 min with MitoSpy-Green FM or MitoSpy-Red CMXRos (Biolegend), diluted in cell culture medium prior to collection. Cells were fixed with 4% PFA and processed by flow cytometry as described above.

### Western blot

Proteins were extracted using the RIPA buffer Lysis System (Santa Cruz). Total protein content was determined in each sample using the Bradford Ultra reagent (Expedeon). Proteins were denatured by heating at 95°C for 5 min and separated by SDS-PAGE. Proteins were subsequently transferred onto PVDF membranes (Immobilon-P, Millipore) and blocked at room temperature with either 5% bovine serum albumin (BSA, Roche), 5% milk or Odyssey buffer

(1:1) diluted in TBST buffer (0.02M Tris-base, 0.15 M NaCl, 0.1% Tween 20, pH 7.4) for 1 h prior to antibody incubation. Membranes were probed with specific primary antibodies in blocking buffer from 16 to 24 h at 4°C. After washing, membranes were subsequently incubated for 2 h at room temperature with the secondary antibodies conjugated with either HRP or Alexa Fluor 680. Secondary antibodies used were donkey anti-rabbit HRP (Alpha Diagnostic International), goat anti-mouse HRP (Sigma-Aldrich), goat anti-rabbit Alexa Fluor 680 or goat anti-mouse Alexa Fluor 680 were from Thermo Fisher scientific. Detection of HRP-conjugated antibodies was carried out the using the Pierce ECL Western Blotting Substrate or the Super-Signal West Femto Maximum Sensitivity Substrate (Thermo Fisher scientific), and images were acquired using ImageQuant LAS 4000 series. Membranes probed with fluorescent antibodies were imaged using the Odyssey Imaging System (LI-COR, Biosciences). Protein signal intensities were quantified using ImageJ (NIH) [33] or the ImageQuant TL 8.1 software (Cytiva).

## Quantitative RT-PCR

Viral RNA was isolated using the RNeasy Mini Kit (Qiagen), following manufacturer's protocol. Absolute viral RNA copies were quantified by RT-qPCR with a standard curve analysis. For this purpose, total RNA was reverse transcribed using the Omniscript RT kit (Qiagen), and the resulting DNA was amplified with the HotStarTaq DNA Polymerase (Qiagen) using sets of primers targeting either the E1 or the nsP1 gene in independent PCR reactions, as described previously [34,35]. For quantification of cellular gene expression, total RNA was obtained using TRIzol (Thermo Fisher scientific) following manufacturer's recommendations. RT-qPCR was carried out using QuantiTect Rev. Transcription Kit (Qiagen), followed by qPCR with Quanti-Nova SYBR Green PCR Kit (Qiagen), using specific primer sets targeting BNIP3 and GAPDH.

## Immunofluorescence

U2OS cells grown on glass coverslips (#1.5 thickness) were fixed with 4% PFA and permeabilized either with 0.1% Triton X-100 in PBS or were simultaneously permeabilized and blocked using blocking buffer (PBS, 1% BSA, 0.1% saponin). Triton X-100 permeabilized cells were separately blocked with 2% BSA. Next, cells were stained for CHIKV infection using the anti-E2 stem or anti-nsP2 antibodies and mitochondria using anti-Tom20 antibody, followed by secondary antibody staining.

For the mitophagy assay, COX8-EGFP-mCherry-expressing U2OS cells were infected with CHIKV at MOI 50 and processed for immunofluorescence. At 4, 6, 8 and 10 hpi cells were washed with PBS and fixed using 3.7% PFA prepared in 200 mM HEPES, pH 7, for 10 min. Then, samples were washed twice with DMEM supplemented with 10 mM HEPES, pH 7.0, followed by a 10 min incubation. Cells were subsequently stained with the anti-CHIKV E2 antibody without permeabilization since detergent can affect the stability of red-only puncta [36].

In all cases, cells were mounted with ProLong Gold Antifade mountant with DAPI (Invitrogen). Images were acquired in a DeltaVision microscope (GE Healthcare Life Sciences) and at least 50 cells per experiment were examined. To determine BNIP3 subcellular localisation, mitochondria were labelled with anti-Tom20 (green) and co-localisation with mCherry-BNIP3 (red) was assessed using Pearson's correlation coefficient. For the mitophagy assay, cells were examined for red-only puncta representing mitolysosomes. All image analysis was performed using the Fiji-ImageJ software (NIH) [33].

## Microscopic membrane hemifusion assay

U2OS cells were reverse-transfected as described above and seeded in Nunc Lab-Tek II Chambered Coverglass (Thermo Fisher scientific) at a density of 9,000 cells per well. The fusion

assay was performed at 48 hpt, using purified CHIKV labelled with the lipophilic fluorescent probe 1,1'-dioctadecyl-3,3,3',3'-tetramethylindodicarbocyanine, 4-chlorobenzenesulfonate (DiD, Invitrogen), as previously described [10]. Prior to infection, cells were washed three times with serum- and phenol red-free MEM (Gibco) and then kept in phenol red-free MEM with 1% of glucose. DiD-labelled CHIKV was added, and cells were incubated for 20 min at 37°C to allow virus endocytosis and membrane fusion. As a control, diethylpyrocarbonate (DEPC; Sigma-Aldrich) treated DiD-labelled CHIKV was used, as described before [10]. At 20 min, unbound virus was removed by washing the cells three times with serum- and phenol red-free MEM. Cells were kept in serum- and phenol red-free MEM containing 1% glucose during the analysis (Leica Biosystems 6000B microscope). A total of 15 random snapshots (55–70 cells in total) were taken per condition, and images were scored by eye for hemifusion-positive cells.

### Statistical analysis

All data were analysed using GraphPad Prism version 9.3.1 for Windows (GraphPad Software). Data are presented as mean ± SEM, unless otherwise stated. Statistical differences were determined using the two-tailed student's $t$-test or the dependent $t$-test, and $p < 0.05$ was considered statistically significant.

## 3. Results

### An ATG proteome-specific siRNA screen identifies BNIP3 as a negative modulator of CHIKV infection

To assess the role of the ATG proteome in CHIKV infection and replication, an unbiased image-based siRNA screen was performed in U2OS cells. U2OS cells are derived from a human osteosarcoma and have been successfully used for studying multiple aspects of the CHIKV replication cycle [37–39]. For the screen, we used a customized siRNA library consisting of 50 different siRNAs targeting individual *ATG* genes and autophagy receptors (S1 Table). Some of these siRNA probes were additionally assorted, such that their combinations silenced functionally redundant proteins (S1 Table). The siRNAs were reverse transfected in U2OS cells [24].

For the screen, we employed a GFP-reporter CHIKV [31] to assess virus infection and replication on the basis of GFP expression. Images were acquired using an automated microscope and the GFP signal was analysed to determine 1) the number of infected cells (% of GFP-positive cells), and 2) the GFP intensity per cell averaged by well (GFP intensity). To define the optimal infection conditions for the screen, we first determined the number of infected cells/ GFP fluorescence intensity over time following infections at different MOIs. S1A and S1B Fig show, respectively, the percentage of infection after each time-point (% GFP-positive cells) and the calculated signal-to-noise ratio based on the mean fluorescence intensity. Infection with MOI 10 resulted in a percentage of infection of 23.9 after 17 hpi and a signal-to-noise ratio of ~7.7. These conditions were therefore chosen for the siRNA screen, since they allowed to properly differentiate specific GFP signal from background noise and to monitor inhibition as well as enhancement of virus infection. In the screen, the percentage of GFP-infected cells in four independent replicates in cells transfected with the siScramble control was 26.5±2.9%. Furthermore, data analysis of the screen results revealed that the cell number in the transfected wells were comparable to those observed in the control cells, indicating good tolerability and no cytotoxicity of both the virus and the siRNA probes (S2 Fig). To analyse the effect of each siRNA, the data of each replica was normalized against the siScramble-transfected cells and the Z-score values were computed for each parameter to determine the statistical significance (Fig 1A and 1B). The siRNA targeting ATP6V1A, a subunit of the lysosomal ATPase, was used

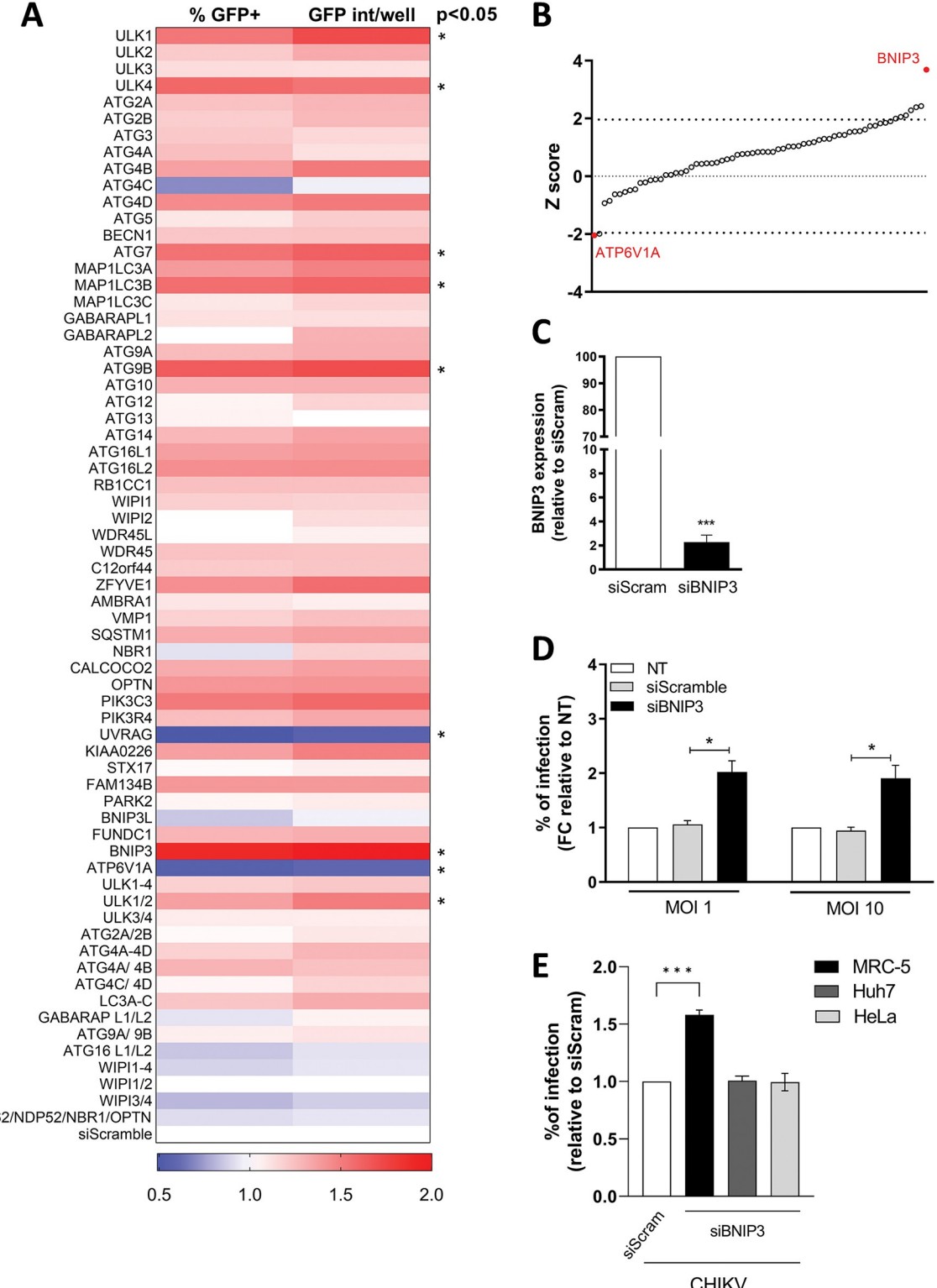

**Fig 1. siRNA-based screen to determine the relevance of ATG proteins and autophagy receptors in CHIKV-infected U2OS cells and validation of BNIP3 as a factor controlling infection. (A)** Heat-map showing the results of the siRNA screen (n = 4). Data represent the ratio of GFP-positive cells (%GFP+) and the average GFP intensity per well (GFP int/well), for each siRNA knockdown as compared to cells transfected with a siRNA control (siScramble, bottom). The significant hits ($p < 0.05$ in at least one of the parameters) are indicated with an asterisk. **(B)** Distribution of the Z-score values of the siRNA screen as compared to the siScramble

based on the percentage of GFP-positive cells. Z-scores between -1.96 and +1.96 represent significant hits with $p$ <0.05. BNIP3 and ATP6V1A Z-scores are indicated in red. **(C-D)** U2OS cells were reverse-transfected in 24-well plates with siRNAs targeting BNIP3 (siBNIP3) or with siScramble (siScram) for 48 h. **(C)** Bar plot showing BNIP3 transcript levels relative to the siScramble as measured by RT-qPCR. **(D)** Transfected cells were infected with the 5'GFP-CHIKV-LR strain at the indicated MOI. Cells were collected and fixed at 10 hpi, and infection (i.e., GFP-positive cells) was determined by flow cytometry. Bar plot shows the percentage of infection relative to non-transfected (NT) cells. **(E)** MRC-5, HeLa or HuH-7 cells were reverse-transfected with siBNIP3 or siScram for 48 h and infected at MOI 5 (HeLa/HuH-7 cells) or MOI 1 (MRC-5 cells). After 9–10 hpi, infection was evaluated as stated in 1D. Data represents mean ± SEM of at least three independent experiments. Student's test: *** $p < 0.001$, * $p < 0.05$.

as a positive control for the screen. ATP6V1A is responsible for endosome acidification and silencing of this protein prevents CHIKV infection [40]. Indeed, ATP6V1A depletion was found to significantly decrease CHIKV replication (Fig 1A and 1B). The analysis of the screen results revealed that most ATG proteins have a mild role in restricting CHIKV replication since their knockdown resulted in an enhanced number of infected cells and/or virus replication (higher GFP intensity) (Fig 1A). A statistically significant effect was observed for 7 individual ATG proteins and ULK 1/2, that is, their Z-score value was below -1.96 or above 1.96 (Fig 1A and 1B). Depletion of UVRAG significantly inhibited CHIKV infection (Fig 1A) whereas individual depletion of 6 different proteins (ULK1, ULK4, ATG7, LC3B, ATG9B and BNIP3) significantly enhanced infection. Concomitant depletion of ULK1 and ULK2 also significantly favoured CHIKV infection. Out of the individual hits, BNIP3, had the most pronounced effect with a 1.9 and 2.4-fold increase in the percentage of infected cells and GFP intensity per well, respectively, compared to the siScramble control.

Given the striking effect of BNIP3 downregulation in comparison to the other ATG proteins, the effect of this protein in CHIKV infection was further investigated. To this end, the knockdown efficiency of the siRNAs used in the primary screen was determined by measuring the mRNA levels of BNIP3 by RT-qPCR in siScramble- versus siBNIP3-transfected cells. The siRNAs targeting BNIP3 reduced BNIP3 mRNA levels by more than 95% in comparison to the siScramble-transfected cells (Fig 1C). Attempts to verify BNIP3 knockdown at the protein level using commercially available antibodies were unsuccessful, a common issue reported in the literature [41,42]. As an alternative approach to verify BNIP3 protein downregulation by siBNIP3, cells transiently overexpressing a BNIP3-mCherry fusion protein were reverse transfected with siScramble or siBNIP3 before measuring BNIP3-mCherry fluorescence by flow cytometry and western blot analysis at 16 h post-transfection using an anti-mCherry antibody. As an additional control, cells were transfected with an empty mCherry vector and the above mentioned siRNAs. In agreement with the RT-qPCR data, transfection with the siRNA targeting BNIP3 led to a strong reduction in both the number of cells positive for BNIP3-mCherry (S3A Fig) and the expression levels of the tagged protein (S3B Fig); confirming that BNIP3 is indeed downregulated by the siRNAs used in this study.

In the screen, the role of ATG proteins on CHIKV infection was assayed at 17 hpi, which represents two rounds of virus replication. To validate the primary screen results and to investigate whether BNIP3 controls CHIKV infectivity in the first round of replication, we next assessed the number of infected cells by flow cytometry at an earlier time-point, i.e., 10 hpi, using MOI 1 and MOI 10. Similar to the siRNA screen results at 17 hpi, knockdown of BNIP3 expression increased the number of infected cells by 2.0- and 1.9-fold upon infection at MOI 1 and 10, respectively (Figs 1D and S3C and S2 Table), in comparison to the siScramble-transfected cells. These results confirm the role of BNIP3 in CHIKV infection and indicate that this protein controls CHIKV infectivity within one round of replication. To further validate the effect of BNIP3 on CHIKV infectivity we next investigated the role of BNIP3 depletion on the number of CHIKV-infected MRC-5, HeLa and HuH-7 cells. Upon BNIP3 depletion, a significant enhancement of the number of infected cells was observed in MRC-5 cells whereas no

effect was observed in HeLa and HuH-7 cells (Fig 1E and raw data in S3 Table). This shows that the effect of BNIP3 in CHIKV infectivity is not limited to U2OS cells yet there is some level of cell type specificity.

## BNIP3 depletion enhances CHIKV infection through an autophagy-independent mechanism

Next, we systematically assessed the interplay between BNIP3 and autophagy to better comprehend the role of BNIP3 in CHIKV infection.

First, we examined whether an autophagic response is elicited by CHIKV infection in U2OS cells (S4 Fig). To this end, the lipidation of the autophagy marker protein LC3, assessed by western blot, was used as a readout of autophagic activity, since lipidated LC3, i.e., LC3-II, tightly associates with nascent autophagosomal membranes [43]. Given that autophagy is a highly dynamic process, the mere detection of LC3 processing at steady state is insufficient to determine autophagic flux [44]. Therefore, the ratio LC3-II/LC3-I in the absence and the presence of the lysosomal inhibitor BafA1, also referred to as autophagy flux index, was evaluated in CHIKV-infected cells [44]. Nutrient deprivation, used as the positive control for autophagy induction, led to an increase in the autophagic flux as expected (S4A and S4B Fig). In contrast, no differences in the autophagic flux were observed during CHIKV infection from 5 to 16 hpi, when compared to mock-treated cells (S4A and S4B Fig), which suggests that autophagy is not induced by CHIKV in U2OS cells. These findings were confirmed in U2OS cells stably expressing GFP-WIPI2 after 5, 10 and 16 h of CHIKV infection. GFP-WIPI2 puncta, which corresponds to nascent autophagosomes, are formed once an autophagic response is triggered [45]. Accordingly, >42.4% of cells showed at least one GFP-WIPI2 puncta upon nutrient starvation (S4C and S4D Fig). In contrast, the percentage of cells that showed GFP-WIPI2 puncta at 5 h, 10 h and 16 h, was 3.4%, 4.6% and 1.0%, respectively, in mock-treated conditions (S4C and S4D Fig). In agreement with the LC3 western blot analysis, no significant differences were seen in the number of cells displaying GFP-WIPI2 puncta exposed to CHIKV at the evaluated time-points (S4C and S4D Fig). These results allow us to conclude that CHIKV infection does not provoke an autophagic response in U2OS cells and thus it raises the question of whether BNIP3 antiviral function relies on autophagy.

BNIP3 is an autophagy receptor specifically involved in mitophagy and not in bulk autophagy [46]. To confirm this notion, we evaluated the effect of BNIP3 depletion on basal autophagic flux in U2OS cells in the absence and the presence of BafA1. Western blot analysis revealed that there were no significant differences in the autophagic flux between siScramble- and siBNIP3-transfected cells (Fig 2A). This result confirms that BNIP3 is not required for basal autophagy (Fig 2A). Subsequently, we assessed whether BNIP3 depletion affects the autophagic flux in CHIKV-infected cells. Fig 2B shows that there are no differences in autophagic flux between siScramble- and BNIP3-depleted cells infected for 10 h with CHIKV at MOI 10, suggesting that the antiviral function of BNIP3 during CHIKV infection does not involve an alteration of the host autophagic flux. Quantification of LC3-II (normalised against GAPDH) in the presence or absence of BafA1 in infected and mock-treated cells retrieved similar results (S4E Fig). To rule out any contribution of canonical autophagy to the antiviral role of BNIP3, we investigated the effect of ATG7 silencing, on CHIKV infection in combination or not with BNIP3 knockdown. We hypothesised that if ATG7-mediated canonical autophagy is not required for BNIP3 role in CHIKV infection, concomitant depletion of both proteins would result in an additive effect. In agreement with our screen data (Fig 1), siRNA-mediated depletion of ATG7 (Fig 2C) enhanced CHIKV infection (Fig 2D and raw data in S4 Table.), albeit to a lower extent than BNIP3. Remarkably, the combined depletion of ATG7 and BNIP3

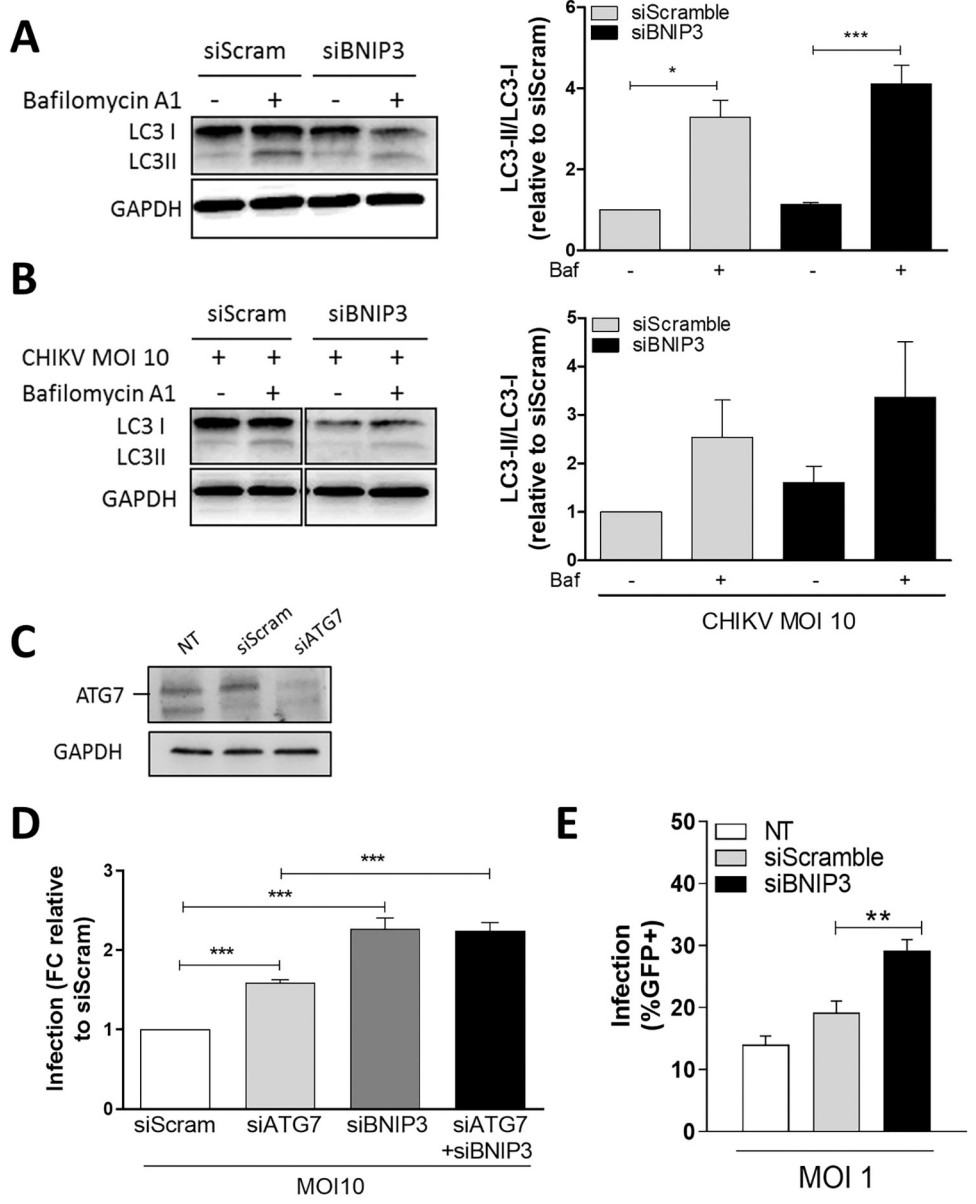

**Fig 2. BNIP3 modulates CHIKV replication independently of autophagy. (A, B, C)** Representative LC3 western blots and determination of the autophagic flux index (LC3-II/LC3-I ratio) in protein lysates from U2OS cells either **(A)** reverse-transfected with siBNIP3 or siScramble for 48 h; **(B)** transfected and infected with 5'GFP-CHIKV-LR at MOI 10 for 10 h. **(C)** Representative ATG7 western blot from protein lysates of U2OS cells reverse-transfected with an siRNA targeting ATG7 (siATG7) or siScramble for 48 h. **(D)** Bar plot showing the percentage of infection relative to siScramble-transfected cells in U2OS cells reverse-transfected for 48 h with either siATG7, siBNIP3, or both, and infected with 5'GFP-CHIKV-LR at MOI 10 for 10 h. Infection was assessed by flow cytometry. **(E)** Bar plot showing the percentage of infection as assessed by flow cytometry of ATG7KO cells reverse-transfected for 48 h with either siBNIP3 or siScramble, and infected with 5'GFP-CHIKV-LR at MOI 1 for 10 h. FC denotes for fold change. Data represents mean ± SEM of at least three independent experiments. NT denotes for non-transfected, and Baf for Bafilomycin $A_1$. Student's test: *** $p < 0.001$, ** $p < 0.01$, * $p < 0.05$, no symbol implies non-statistically significant.

further increased CHIKV infectivity when compared to siATG7-transfected cells alone (Fig 2D). Similarly, in CRISPR/Cas9-generated *ATG7* knockout U2OS cells (ATG7KO) [28], silencing of BNIP3 resulted in a significant enhancement of the number of CHIKV-infected

cells in comparison to the ATG7KO cells transfected with the siScramble control (Fig 2E). Altogether, these results indicate that BNIP3 depletion has a positive effect on CHIKV infectivity, even when canonical autophagy is inhibited.

## CHIKV infection does not alter mitophagy

Since BNIP3 specifically participates in mitophagy and in other regulatory mechanisms involved in maintaining the homeostasis of the cellular mitochondrial content [46–48], we next assessed the mitophagic flux in CHIKV-infected cells in the presence or the absence of BNIP3. To this aim, we used U2OS cells stably expressing a construct consisting of COX8, an inner mitochondrial membrane protein, fused to EGFP and mCherry. Since the fluorescence of EGFP, but not that of mCherry, is affected by a low pH; this reporter construct allows discriminating cytoplasmic mitochondria, i.e, EGFP/mCherry-positive ones, from mitochondria localizing to acidic autolysosomes, i.e., red-only ones. Cells expressing COX8-EGFP-mCherry were infected at MOI of 50 to maximize infection and assessed in a time-course manner by fluorescence microscopy after 4, 6, 8 and 10 hpi. Fig 3A shows representative images of the 10 h time-point. As expected [49], quantification of red-only puncta in cells treated with the iron-chelator DFP, a known mitophagy inducer, and transfected with siScramble or siBNIP3, showed a clear increase when compared to mock-treated cells, with an average of 5.8 and 9.9 red-only puncta per cell, respectively (Fig 3B). However, no changes in the mitophagy flux were seen upon CHIKV infection at any of the assessed time-points, irrespective of BNIP3 expression, even when cells depleted for BNIP3 were more susceptible to infection (Fig 3C).

To corroborate these findings and also explore changes in mitochondrial membrane potential ($\Delta\Psi m$) and indirectly mitochondrial function, we next assessed differences in mitochondrial mass and $\Delta\Psi m$ in BNIP3-depleted cells during CHIKV infection. To this end, the total mitochondrial mass and the mass of polarized mitochondria were quantified by flow cytometry using specific probes. Starvation-induced increase in mitochondrial mass as a consequence of enhanced mitochondrial fission [50] was used as a positive control for the assay (S5A Fig). Overall, no changes in the total mitochondrial content and $\Delta\Psi m$ were detected upon CHIKV infection (S5B and S5C Fig), regardless of the presence of BNIP3. This data further confirms that BNIP3 function in maintaining mitochondrial homeostasis is not required for its antiviral role in CHIKV-infected cells.

Lastly, we infected U2OS cells stably expressing BNIP3-mCherry with CHIKV to determine whether this virus affects BNIP3 localisation. As shown in Fig 3C, BNIP3-mCherry colocalised with mitochondria under all the tested conditions (Pearson correlation coefficient >0.8), indicating that BNIP3 remains associated to mitochondria during CHIKV infection.

Altogether, these results underline that BNIP3 controls CHIKV independently of its functions in mitophagy and mitochondrial homeostasis.

## The pro-apoptotic function of BNIP3 is not required for CHIKV infection

BNIP3 expression has been shown to reduce the $\Delta\Psi m$ and increase the generation of reactive oxygen species, thereby activating apoptosis and other types of cell death in specific cellular models [51]. Since CHIKV infection is known to induce apoptosis [52], we next explored whether BNIP3 silencing influences CHIKV-induced cell death. We evaluated cell death by flow cytometry using a protocol based on Annexin V and the fluorescent viability dye FVD (Fig 4A). This double-sensor approach allows discriminating three distinct populations of dying cells: 1) early apoptotic cells (Annexin V-positive); 2) apoptotic cells (positive for both Annexin V and FVD); and 3) necrotic cells or cells undergoing other cellular death programmes (FVD-positive cells) [52]. As a positive control, U2OS cells were treated with

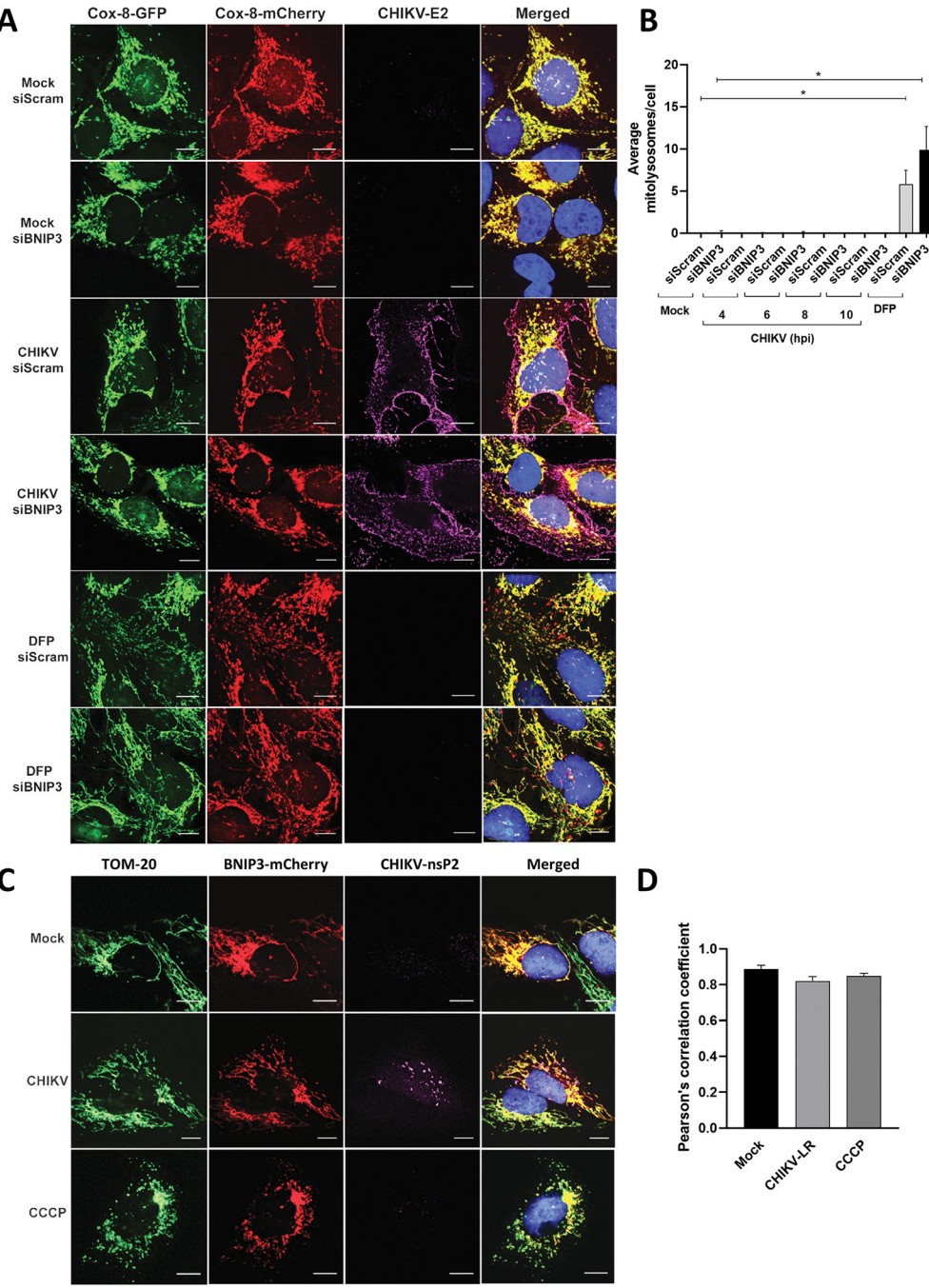

**Fig 3. CHIKV replicates independently of mitophagy. (A)** Representative fluorescence microscopy images, **(B)** quantification of the mitolysosomes/cell and **(C)** percentage of infection in U2OS-Cox-8 GFP-mCherry cells reverse transfected with siBNIP3 or siScramble for 48 h and either mock-treated, infected with CHIKV-LR at MOI 50 for 10 h or treated with DFP at a concentration of 1mM for 24 h. Infection was detected by staining with anti CHIKV-E2 stem antibody. **(D)** Representative fluorescence microscopy images and **(E)** quantification (Pearson's correlation coefficient) of U2OS cells stably over-expressing BNIP3-mCherry and infected with CHIKV-LR at MOI 50 for 10 h or treated with the mitochondrial uncoupler CCCP at a concentration of 10 μM for 10 h. Mitochondrial network and CHIKV infection are detected with antibodies targeting TOM20 and CHIKV-nsP2 antibody. Scale bar 10 μm. Student's test: * $p < 0.05$, no symbol implies non-statistically significant.

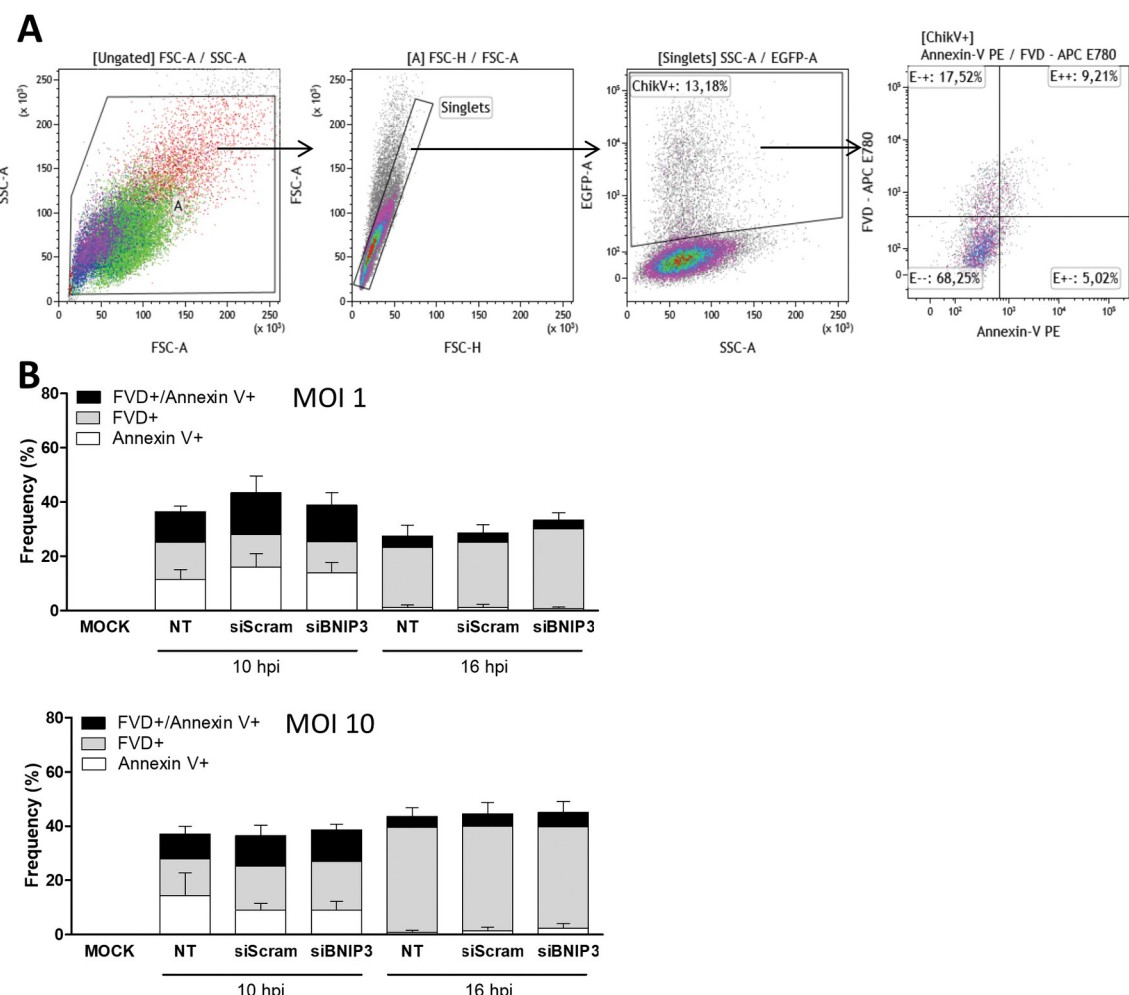

**Fig 4. BNIP3 involvement in CHIKV infectivity is independent of cellular death pathways.** U2OS cells were reverse-transfected with siBNIP3 or siScramble for 48 h. Cells were infected with 5'GFP-CHIKV-LR at the indicated MOI for 10 and 16 h. **(A)** Gating strategy and **(B)** bar plots showing the frequencies of Annexin V-, FVD- and double positive cells measured by flow cytometry in GFP + (infected) cells. Data represents mean ± SEM of at least three independent experiments. NT denotes for non-transfected, siScram for siScramble. Student's test: no symbol implies non-statistically significant.

menadione, which is known to induce apoptosis and other types of cell death [53,54]. In presence of menadione, 36.2% of the total cell population indeed showed signs of cell death (percentage of FVD-positive and Annexin V/FVD double positive cells) (S5D Fig). Furthermore, in agreement with the literature [55,56], we observed a MOI- and time-dependent increase in CHIKV-induced cell death as underlined by the increased percentage of FVD-positive cells over time (Fig 4B). Infection of siScramble-transfected cells with CHIKV at MOI 1 led to 12.0% FVD-positive cells at 10 hpi, which increased to 23.9% at 16 hpi. Infection with MOI 10, led to 16.1% and 38.6% of FVD-positive cells at 10 and 16 hpi, respectively. Furthermore, apoptotic cells were predominantly detected at early time points post-infection at MOI 1 and 10 (Fig 4B). Lastly, Annexin V-positive cells were almost exclusively detected at the early stages of infection (Fig 4B, 10 hpi). BNIP3-depleted CHIKV-infected cells had similar percentages of Annexin V-, FVD- and double positive cells as in the control cells at 10 and 16 hpi, and at both high and low MOI (Fig 4B). Collectively, these results indicate that BNIP3 control of CHIKV replication is not associated to mechanisms related to cell death.

## CHIKV replication is modulated by BNIP3 early in infection

Next, we investigated at which stage of the viral cycle BNIP3 modulates CHIKV. First, the influence of BNIP3 silencing on CHIKV cell entry and membrane hemifusion was determined by live-cell imaging using DiD-labelled CHIKV. Hemifusion refers to a short-lived state in which the outer leaflet of the virion membrane has fused with the acceptor membrane while the inner leaflet is still intact, thus referring to a state prior to fusion, pore formation and viral gRNA release into the cytoplasm [57]. In the assay, non-fusogenic DEPC-inactivated CHIKV was used as a positive control. DEPC treatment inactivates the virus, yet, it preserves the conformational integrity of its surface proteins required for cell surface attachment and endocytosis [58]. As expected, the results show that the percentage of fusion-positive cells was reduced from 36% to 1% when CHIKV was DEPC-inactivated (Fig 5A). BNIP3 depletion, however, did not interfere with virus cell entry and membrane hemifusion, as no differences in the number of fusion-positive cells was observed in comparison with the siScramble-treated cells (Fig 5A). This result indicates that BNIP3 regulates CHIKV infection at a step after virus cell entry and membrane hemifusion.

Upon RNA delivery into the cell cytoplasm, the CHIKV genome is translated to produce non-structural proteins (nsP1-4) after which, the RNA is replicated and a subgenomic viral RNA is translated to produce the structural capsid, E2, E1 and E3 proteins. To establish whether BNIP3 interferes with the translation of the CHIKV proteins, we next assessed the expression levels of the viral proteins by western blot (Fig 5B and 5C) at 10 hpi. BNIP3 silencing was found to increase the expression of the non-structural protein 2 (nsP2) and the structural proteins E2, E1, and the capsid protein (Fig 5B and 5C). Collectively, these data indicate that BNIP3 depletion increases the expression of CHIKV proteins translated from the viral genomic and subgenomic RNA.

We next evaluated the effect of BNIP3 depletion on intracellular viral RNA levels. The number of viral RNA copies was determined by RT-qPCR using primer sets targeting nsP1 and E1. The nsP1 primer set only amplifies the genomic RNA whereas the E1 primer set amplifies both the viral genomic and subgenomic RNA [34,35]. Due to the higher detection sensitivity of RT-qPCR compared to western blot, we analysed viral RNA levels at 4 and 8 hpi. As expected, the total RNA copy number per ml increased with the MOI and over time (S6A Fig). Strikingly, there was a 5.3- and 6.7-fold increase in the number of viral RNA copies of the E1 and nsP1 transcripts, respectively, in BNIP3-depleted cells in comparison to the cells transfected with the siScramble at MOI 1 at 4 hpi (Fig 5D). Similarly, the E1 and nsP1 transcripts were significantly increased when BNIP3 was silenced in cells infected at MOI 10 (Fig 5D). At 8 hpi, the fold change in viral intracellular RNA levels in BNIP3-depleted cells was less prominent, although statistically significant in the case of the E1 transcripts (Fig 5D). Furthermore, no clear differences were observed between the fold enhancement of nsP1 and E1 RNA levels at all conditions tested, suggesting that BNIP3 interferes with the replication of both the viral genomic and subgenomic RNA.

Next, we examined whether the higher viral protein and RNA synthesis observed in BNIP3-deficient cells was resulting in a higher release of viral particles. Hereto, we first analysed the number of genome equivalent copies (GEC) per ml of supernatant secreted by siScramble-transfected or BNIP3-depleted U2OS cells at 10 hpi following CHIKV infection at MOI 1 and 10 (Fig 5E). In siScramble-transfected cells, on average $2.7 \times 10E7$ GEC/ml at MOI 1 and $1.2 \times 10E8$ GEC/ml at MOI 10 were produced at 10 hpi. Consistent with the intracellular viral RNA levels, the western blot data, and the higher number of infected cells (Figs 1D and 5A–5D), the secreted GEC/ml at MOI 1 and 10 increased in BNIP3-depleted cells by 2.6 and 2.7-fold, respectively (Fig 5E). To rule out effects on the infectivity of the secreted viral

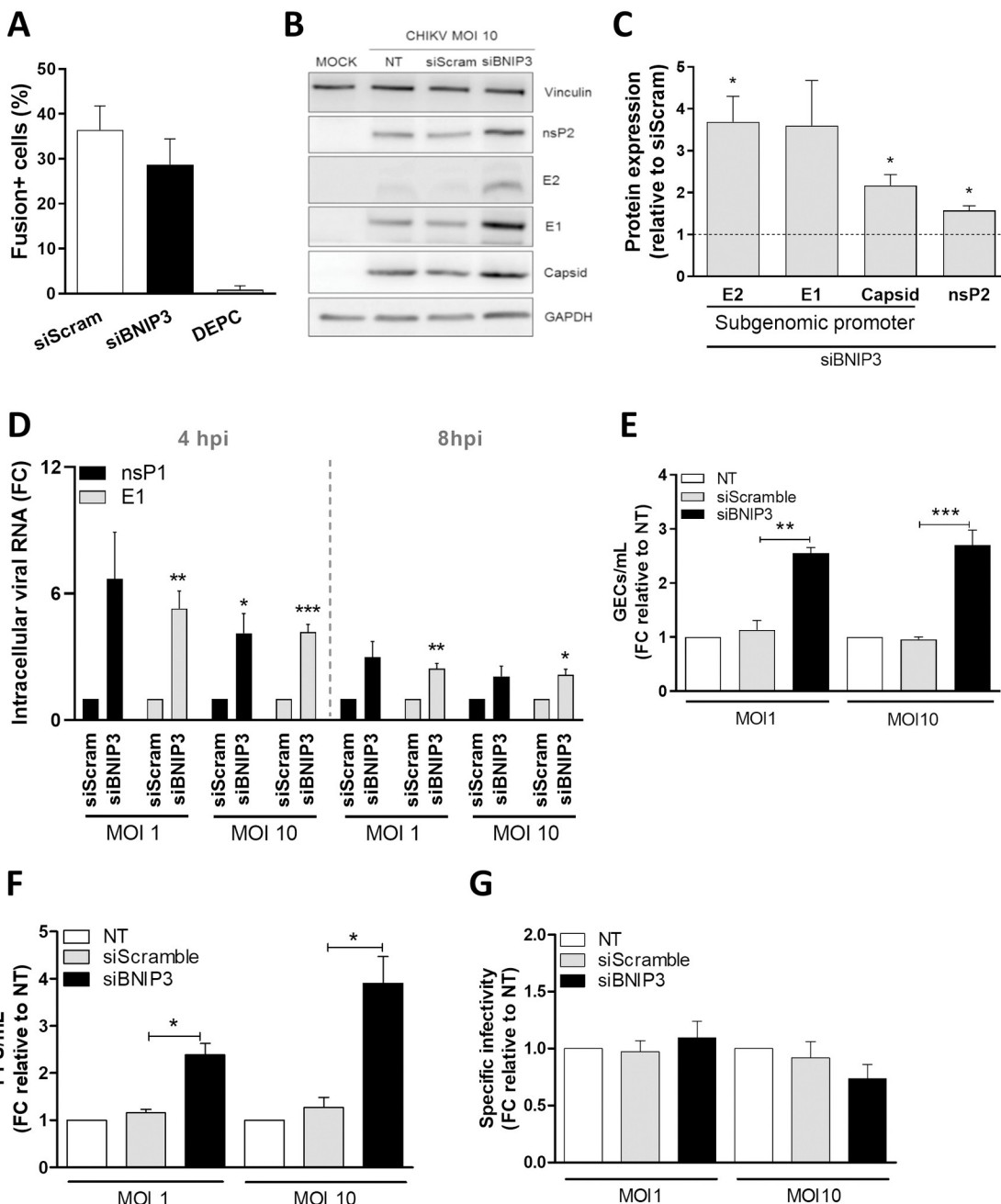

**Fig 5. BNIP3 controls CHIKV replication early in infection.** U2OS cells were reverse-transfected with siBNIP3 or siScramble for 48 h. **(A)** Transfected cells were incubated with DiD-labelled CHIKV-LR and the number of fusion-positive cells was quantified by TIRF microscopy. **(B-G)** siRNA-transfected cells were infected with 5'GFP-CHIKV-LR at the indicated MOI. **(B)** Representative blots and **(C)** bar plot showing the expression levels of CHIKV nsP2, E2, E1 and capsid proteins in BNIP3-depleted cells compared to siScramble-transfected cells after 10 hpi. GAPDH and vinculin were both used as loading controls. **(D)** Bar plot showing the levels of intracellular CHIKV E1 and nsP1 RNA in BNIP3-knockdown cells compared to siScramble-transfected cells. **(E)** Bar plot showing the quantification by RT-qPCR of genome equivalent copies per ml of supernatant (GECs/ml) after 10 hpi, relative to NT cells. **(F)** Bar plot showing the titration of infectious CHIKV particles per ml of supernatant (PFU/ml, determined by plaque assay) after 10 hpi and expressed as relative to NT cells. **(G)** Bar plot showing the specific infectivity (GECs/ml divided by the PFU/ml) compared to NT cells. FC denotes for fold change. Data represents mean ± SEM of at least three independent experiments. Student's test: *** $p < 0.001$, ** $p < 0.01$, * $p < 0.05$, no symbol implies non-statistically significant. NT denotes for non-transfected, siScram for siScramble.

particles, we quantified the infectious CHIKV titres by plaque assay on the same samples (Fig 5F). The titers in siScramble-transfected cells were on average 2.6x10E4 PFU/ml and 2.1x10E5 PFU/ml at MOI 1 and 10, respectively. BNIP3 knockdown led to a 2.2- and 3.8-fold increase in the secretion of infectious virus particles at MOI 1 and 10, respectively, relative to siScramble-transfected cells at 10 hpi (Fig 5F). Similar results were obtained at 16 hpi (S6B Fig). Subsequent calculation of the GEC to PFU ratio revealed no differences (Fig 5G), indicating that the higher percentage of infection observed in BNIP3-depleted cells results in a higher CHIKV titre without altering the infectious properties of the released viral particles. Taken together, these data show that BNIP3 is a host factor that controls CHIKV infection in U2OS cells.

## BNIP3 negatively modulates the replication of other alphaviruses

The CHIKV strain used in this study, i.e., La Reunion (CHIKV-LR), belongs to the Indian Ocean lineage. To address whether BNIP3 also controls the infection of other CHIKV strains, we examined the effect of BNIP3 depletion on the viral cycle of CHIKV-S27, the prototype strain originally isolated in 1953 and CHIKV-99659, a recently isolated Caribbean strain that belongs to the Asian Urban lineage. We assessed both the number of infected cells by flow cytometry (Fig 6A) and the infectious viral particles produced following infection at MOI 10. At 10 hpi, on average 19.7% and 9.7% of the cells were infected with CHIKV-S27 and CHIKV-99659, respectively. At 10 hpi, the infectious virus titres were on average 3.16x10E6 PFU/ml for CHIKV-S27 and 3.3x10E5 for CHIKV-99659. For the ancient CHIKV-S27 strain, we observed a non-significant positive trend with a 1.4- and 2.0-fold increase in the percentage of infected cells and infectious virus particle production, respectively (Fig 6B). For the circulating CHIKV-99659 strain, BNIP3 silencing resulted in a significant increase in both the percentage of infected cells (2.5-fold) as well as the secretion of infectious virus progeny (2.3-fold) (Fig 6B).

Given that SFV is part of the same antigenic complex as CHIKV, we also tested the effect of BNIP3 depletion towards this virus. Transfection with siScramble resulted in on average 24.8% infected cells and 7.62x10E6 PFU/ml after 10hpi. Similar to our observations with CHIKV-S27, BNIP3 silencing had a non-significant yet positive trend on the number of infected cells (1.4-fold increase, Fig 6B). This translated into a significant 1.9-fold enhancement in virus progeny release as compared to the siScramble-transfected cells (Fig 6B). This shows that the role of BNIP3 in regulating CHIKV-LR infection could be involved in the replication of other CHIKV strains and other alphaviruses from the same antigenic complex.

## Discussion

In this study we unveiled BNIP3 as a host factor negatively interfering with alphavirus replication, particularly CHIKV. That is, BNIP3 depletion leads to an increase in the number of infected cells and the production of infectious progeny virus particles (Fig 7). BNIP3 interferes with the early stages of the virus replication cycle, yet after virus cell entry and hemifusion with the endosomal limiting membrane (Fig 7). This role of BNIP3 in CHIKV, however, is not connected with the function of this protein in mitophagy and cellular death.

The siRNA screen data presented here shows that the depletion of most ATG proteins has a moderate yet positive effect on CHIKV infection, corroborating previous evidence indicating an antiviral role of the autophagy machinery [20,59–61]. Indeed, downregulation of core components of the autophagy pathway such as ULK1, ULK4 ATG7, LC3B and ATG9B significantly enhanced CHIKV infection. Simultaneously, our observations also suggest that although autophagic degradation of viral and hosts factors required for virus replication might occur, autophagy is not massively activated after infection. Whether CHIKV is actively

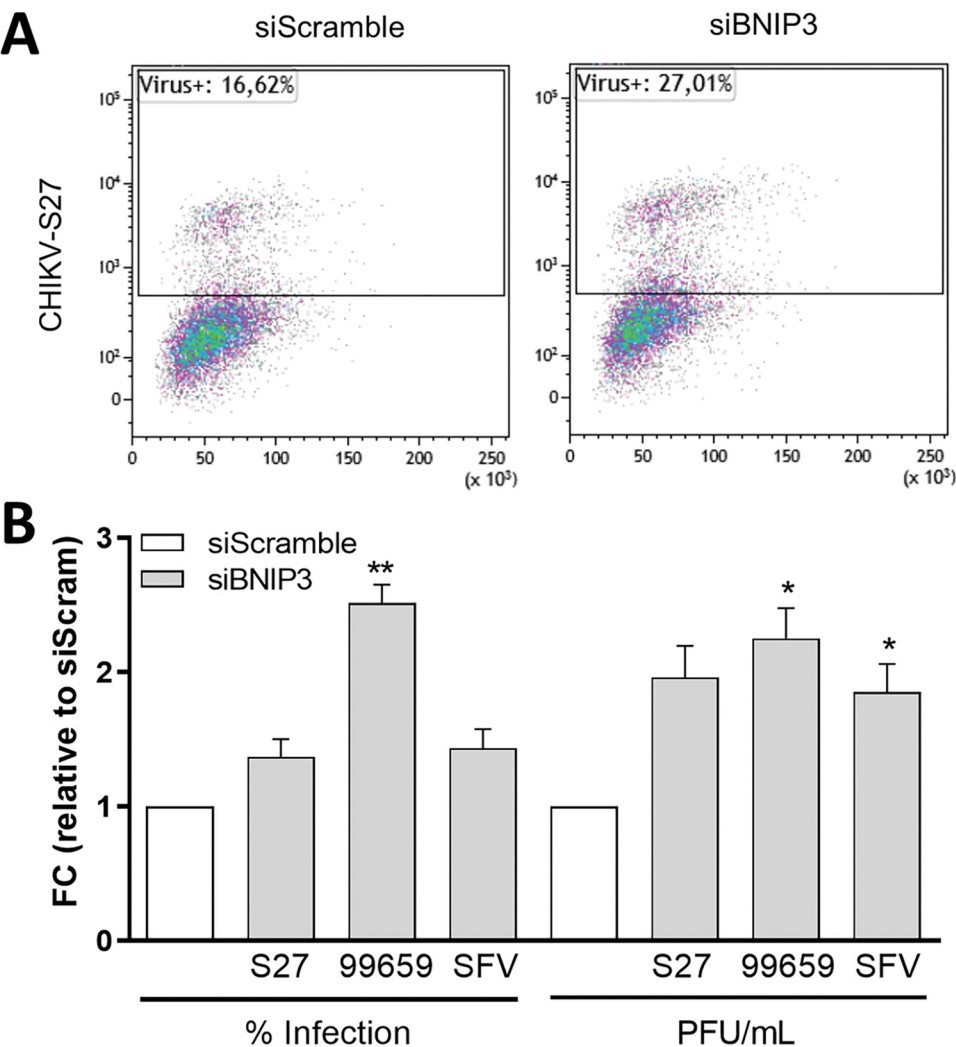

**Fig 6. BNIP3 antiviral role is conserved across CHIKV genotypes and SFV.** U2OS cells were reverse-transfected with siBNIP3 or siScramble for 48 h. Transfected cells were infected at MOI of 10 for CHIKV-S27 and CHIKV-99659 and at MOI 1 for SFV-MK for 10 h. **(A)** Representative flow cytometry dot plot and **(B)** bar plots showing the percentage of E2-positive/infected cells and the number of secreted infectious virus particles per ml (PFU/ml) in BNIP3-depleted cells relative to siScramble-transfected cells. FC denotes for fold change. Data represents the mean ± SEM of at least three independent experiments. Student's test: ** $p < 0.01$, * $p < 0.05$, no symbol implies non-statistically significant.

counteracting autophagy initiation, as observed with other viruses [62,63], remains to be determined.

The protein with the most pronounced effect on CHIKV infection in the primary siRNA screen was BNIP3, a pro-apoptotic mitochondrial protein and autophagy receptor. To validate target downregulation by siRNAs at the protein level usually western blot analysis is performed. For BNIP3, the detection of endogenous levels of the protein using commercial antibodies appeared challenging, as also noted in the literature [41,42]. As an alternative approach, we validated the knockdown efficiency of siBNIP3 using cells transiently overexpressing mCherry-BNIP3. Next to a marked reduction in protein expression, the transcript levels of endogenous BNIP3 were also found reduced up to 95% after siRNA transfection, demonstrating the efficiency of the knockdown and validating the screen results.

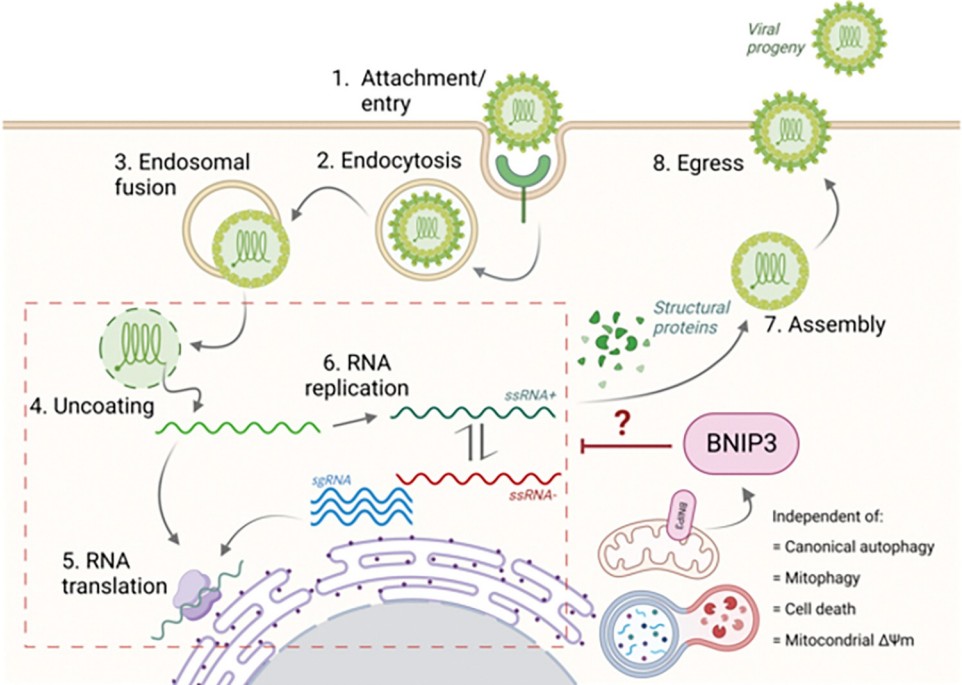

**Fig 7. Schematic representation of CHIKV replication cycle and the involvement of BNIP3 in the early stages of replication.**

Due to its function in autophagy-mediated clearance of damaged mitochondria, BNIP3 is classified as a mitophagy receptor [64]. Previously, other autophagy receptors such as p62 and SMURF1 were associated with the control of alphavirus infection [60,65,66]. Mechanistically, p62 and SMURF1 mediate the autophagic degradation of the alphavirus capsid protein [60,65,66]. BNIP3 can regulate autophagy via its interaction with the mTOR activating GTPase, Rheb, leading to mTORC1 inactivation and thereby triggering autophagy via ULK1 and the downstream ATG machinery [67]. The novel role of BNIP3 reported here, however, does not rely on the ATG machinery, as ATG7 was not essential for this regulatory function. Our findings add to the mounting evidence indicating that autophagy receptors participate in cellular processes other than autophagy, including inflammation, amino acid sensing, and the oxidative stress response [68]. In this regard, it was recently reported that BNIP3 upregulates the production of inflammatory mediators such as TNF-α, MCP-1, IL-6 and IL-1β in enterovirus 71-infected cells [69]. Future research should focus on exploring whether this immunomodulatory function of BNIP3 plays a role in CHIKV, especially since there is a well-established link between the production of inflammatory mediators such as MCP-1 and IL-6, and the pathogenesis and chronic manifestations caused by CHIKV and other arthritogenic alphaviruses [70–73].

The involvement of BNIP3 in the pathogenesis of diseases such as heart failure and cancer has been extensively studied before and it is associated with the regulation of mitochondrial turnover and cell death programs like apoptosis and necrosis. These functions of BNIP3, however, are not likely to be responsible for the attenuation of CHIKV replication described here. Indeed, quantification of dead cell populations in infected cells revealed comparable data between control and BNIP3-depleted cells. We did not directly study the induction of necrosis yet the viability dye used in the flow cytometry experiments has the ability to stain non-viable

cells irrespectively of the cell death mechanism. A few studies have reported connections between BNIP3 and the receptor-interacting protein kinase-3 (RIPK3), underlying the participation of mitochondrial events in the process of necroptosis [74,75]. To completely rule-out the role of necroptosis in the function of BNIP3 in CHIKV life cycle, potential interactions between BNIP3 and RIPK3 should be investigated. BNIP3 expression has also previously shown to lower ΔΨm and cellular ATP levels by reducing oxidative phosphorylation and mitochondrial calcium levels [76,77]. Our measurements did not detect changes in the ΔΨm after CHIKV infection and BNIP3 depletion. Although the ATP levels in CHIKV-infected cells were not assessed in this study, it has been reported that the cellular ATP content is not affected by sindbis virus, another member of the alphavirus genus [78].

An astonishing observation was that BNIP3 silencing increased CHIKV RNA levels as early as 4 hpi. Intriguingly, no major effect was detected on virus cell entry and hemifusion. We therefore conclude that BNIP3 possibly interferes with the stages occurring after the nucleocapsid is delivered into the cytoplasm and prior to robust RNA replication and translation. Early alphavirus RNA synthesis starts in spherules located on the plasma membrane, which are then internalized in an actin-myosin dependent manner and form cytopathic vacuoles, where RNA replication continues [79,80]. Therefore, an intact actin network is found to be essential for the trafficking of alphavirus replication complexes [81]. Conversely, at later time-points after infection, alphaviruses induce the disruption of the actin microfilaments [81,82]. In this regard, loss of BNIP3 has been shown to cause the polymerization of actin into bundles (stress fibres) and to increase F-actin staining in melanoma cells [83]. Thus, studying the cytoskeleton organization in BNIP3-depleted cells infected with CHIKV might deepen our understanding of the molecular events involved in the mobilisation and maturation of alphavirus replication spherules and cytopathic vacuoles.

In sum, our study identified BNIP3 as a novel host factor regulating alphavirus infections. BNIP3 controls CHIKV infection at an early step in its replication cycle. This regulation is not linked to the known function of BNIP3 in autophagy and cellular death pathways. Future studies on how BNIP3 controls alphavirus infection will shed light onto both new cellular functions of this protein and the early steps of the alphavirus replicative cycle, which still remain poorly characterized.

## Supporting information

**S1 Fig. Time-course of U2OS cells infection with CHIKV. Cells were infected with either MOI 1, 5 or 10 of 5'GFP-CHIKV-LR, fixed and stained with Hoechst at the indicated time points after infection. Images were acquired in a Cellomics ArrayScan VTI HCS Reader.** **(A)** Percentage of GFP+ (infected) cells after each time-point. **(B)** Bar plot showing the signal-to-noise ratio after each time point, i.e, the mean fluorescence intensity (MFI) from infected wells (signal) divided by the MFI of the corresponding mock (noise).
(DOCX)

**S2 Fig. siRNA transfection does not induce cell toxicity. (A)** Plot shows the number of cells for each gene knockdown in the siRNA-based screen presented in Fig 1A and 1B. Data represents the mean ± SEM of four independent experiments.
(DOCX)

**S3 Fig. siRNA-mediated knockdown of BNIP3 and validation.** U2OS cells transiently expressing mCherry-BNIP3 were reverse-transfected with either siScramble or siBNIP3 for 16 h. **(A)** Representative flow cytometry dot plot showing the percentage of mCherry-BNIP3-positive cells. **(B)** Representative blot and bar plot showing the mCherry protein expression by

western blot. **(C)** Representative fluorescent micrographs taken with a 10X magnification objective of U2OS cells infected with MOI 10 from Fig 1D, prior to collection for flow cytometry. FC denotes for fold change. Data shown represents the mean ± SEM of at least three independent experiments. Student's test: ** $p < 0.01$, no symbol implies non-statistically significant.
(DOCX)

**S4 Fig. CHIKV does not trigger an autophagic response in U2OS cells. (A)** Representative LC3 western blots and **(B)** determination of the autophagic flux index (LC3-II/LC3-I ratio) in protein lysates from U2OS cells either mock-treated or infected with CHIKV-LR at MOI 10 at the indicated time-points. **(C)** Representative fluorescence microscopy images and **(D)** quantification of GFP-WIPI2 puncta in U2OS cells stably expressing GFP-WIPI2 and infected with CHIKV-LR at MOI 10. Cells were analysed at the indicated time-points. **(E)** Data presented in Fig 2A and 2B was quantified and shown as LC3-II normalised to the housekeeping gene (GAPDH). In all cases, starvation-induced autophagy is used as a positive control (HBSS/EBSS). Baf, Bafilomycin $A_1$. Data shown represent mean ± SEM of at least three independent experiments. Student's test: *** $p < 0.001$, ** $p < 0.01$, no symbol implies non-statistically significant.
(DOCX)

**S5 Fig. Determination of mitochondrial mass and cell death in U2OS cells by flow cytometry. (A)** Representative flow cytometry histogram of U2OS cells treated with HBSS for 10 h to induce mitochondrial fission and probed using MitoSpy Green. **(B-C)** U2OS cells were reverse-transfected with siBNIP3 or siScramble for 48 h and infected with CHIKV-LR for 10 h at the indicated MOI and stained with mitochondrial probes. Bar plots show **(B)** the total mitochondrial mass (MitoSpy Green FM), or **(C)** the mass of polarized mitochondria (MitoSpy Red CMXRos), relative to mock-treated cells. **(D)** Representative flow cytometry dot plot and quantification of U2OS cells treated with menadione at a concentration of 100 μM for 10 h and stained using Annexin V and FVD. NT denotes for non-transfected. FC denotes for fold change. Data shown represent mean ± SEM of at least three independent experiments. Student's test: no symbol implies non-statistically significant.
(DOCX)

**S6 Fig. CHIKV RNA levels and virus titres are enhanced by BNIP3 silencing.** U2OS cells were reverse-transfected with siBNIP3 or siScramble for 48 h, and infected with CHIKV-LR for 10 h. **(A)** Bar plot showing the intracellular levels of E1 and nsP1 RNA (expressed as RNA copies/ml of cell lysate) in cells transfected with siScramble and infected at the indicated MOI and time-points. **(B)** Bar plot showing the titration of infectious CHIKV particles per ml of supernatant (PFU/ml, determined by plaque assay) after 16 hpi and expressed as relative to NT cells. NT denotes for non-transfected. FC denotes for fold change. Data shown represent mean ± SEM of three independent experiments. Student's test: *** $p < 0.001$, ** $p < 0.01$, * $p < 0.05$, no symbol implies non-statistically significant.
(DOCX)

**S1 Table. Genes targeted by the customised siRNA library and gene accession numbers.**
(DOCX)

**S2 Table. Raw data belonging to Fig 1D.**
(DOCX)

**S3 Table. Raw data belonging to Fig 1E.**
(DOCX)

**S4 Table. Raw data belonging to Fig 2D.**
(DOCX)

## Acknowledgments

The authors thank Izabela Rodenhuis-Zybert for her comments and suggestions during the study, and D. Chan, M. Diamond, S. Guenther, M. Kielian, I. Novak and G. Pijlman for reagents.

## Author Contributions

**Conceptualization:** Liliana Echavarria-Consuegra, Nilima Dinesh Kumar, Marleen van der Laan, Mario Mauthe, Fulvio Reggiori, Jolanda M. Smit.

**Data curation:** Nilima Dinesh Kumar, Marleen van der Laan, Fulvio Reggiori, Jolanda M. Smit.

**Formal analysis:** Liliana Echavarria-Consuegra, Nilima Dinesh Kumar, Marleen van der Laan, Mario Mauthe, Denise Van de Pol.

**Funding acquisition:** Liliana Echavarria-Consuegra, Fulvio Reggiori, Jolanda M. Smit.

**Investigation:** Liliana Echavarria-Consuegra, Nilima Dinesh Kumar, Marleen van der Laan, Mario Mauthe, Denise Van de Pol, Fulvio Reggiori, Jolanda M. Smit.

**Methodology:** Liliana Echavarria-Consuegra, Nilima Dinesh Kumar, Marleen van der Laan, Mario Mauthe, Denise Van de Pol.

**Project administration:** Liliana Echavarria-Consuegra, Nilima Dinesh Kumar, Denise Van de Pol.

**Resources:** Liliana Echavarria-Consuegra, Nilima Dinesh Kumar.

**Supervision:** Mario Mauthe, Fulvio Reggiori, Jolanda M. Smit.

**Validation:** Liliana Echavarria-Consuegra, Nilima Dinesh Kumar, Denise Van de Pol.

**Visualization:** Liliana Echavarria-Consuegra, Nilima Dinesh Kumar, Denise Van de Pol, Fulvio Reggiori, Jolanda M. Smit.

**Writing – original draft:** Liliana Echavarria-Consuegra, Fulvio Reggiori, Jolanda M. Smit.

**Writing – review & editing:** Liliana Echavarria-Consuegra, Nilima Dinesh Kumar, Marleen van der Laan, Mario Mauthe, Denise Van de Pol, Fulvio Reggiori, Jolanda M. Smit.

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
