## [Decision Letter · Decision Letter 0]

24 Oct 2022

Dear Dr. Smit,

Thank you very much for submitting your manuscript "BNIP3 regulates Chikungunya virus infection independently of autophagy and cell death" for consideration at PLOS Neglected Tropical Diseases. As with all papers reviewed by the journal, your manuscript was reviewed by members of the editorial board and by several independent reviewers. In light of the reviews (below this email), we would like to invite the resubmission of a significantly-revised version that takes into account the reviewers' comments. 

We cannot make any decision about publication until we have seen the revised manuscript and your response to the reviewers' comments. Your revised manuscript is also likely to be sent to reviewers for further evaluation.

Sincerely,

Gregory Gromowski

Academic Editor

Abdallah Samy

Section Editor

Editor Comments: I think your manuscript requires careful revision to consider all comments raised in the review process, particularly all points raised by reviewer #2. I hope you will be able to address all these points before submitting your revision. 

Reviewer's Responses to Questions

**Key Review Criteria Required for Acceptance?**

**Methods**

-Are the objectives of the study clearly articulated with a clear testable hypothesis stated?

-Is the study design appropriate to address the stated objectives?

-Is the population clearly described and appropriate for the hypothesis being tested?

-Is the sample size sufficient to ensure adequate power to address the hypothesis being tested?

-Were correct statistical analysis used to support conclusions?

-Are there concerns about ethical or regulatory requirements being met?

Reviewer #1: (No Response)

Reviewer #2: (No Response)

**Results**

-Does the analysis presented match the analysis plan?

-Are the results clearly and completely presented?

-Are the figures (Tables, Images) of sufficient quality for clarity?

Reviewer #1: (No Response)

Reviewer #2: (No Response)

**Conclusions**

-Are the conclusions supported by the data presented?

-Are the limitations of analysis clearly described?

-Do the authors discuss how these data can be helpful to advance our understanding of the topic under study?

-Is public health relevance addressed?

Reviewer #1: (No Response)

Reviewer #2: (No Response)

**Editorial and Data Presentation Modifications?**

Reviewer #1: (No Response)

Reviewer #2: (No Response)

**Summary and General Comments**

Reviewer #1: Flawless work from all points of view:

Methodological rigor, syntax, clarity in the presentation, and presentation of the results. Considerable discussions with comparisons with the literature.

Few minor points to address:

- Please describe how the experimental conditions (MOI, hpi) of viral infection were established. Are they determined by preliminary experiments or inferred from literature? 

- In autophagic flux experiments, the expression levels of LC3-II in presence and absence of the autophagic inhibitor/stimulator should be preferred instead of LC3-II/LC3-I ratio.

- To avoid off target effects, is it possible to evaluate the autophagic flux in infected cells treated with other inhibitors (such as chloroquine) and activators (rapamycin)?

- Similarly, to avoid cell-specific effects, representative experiments (for example, the validation of BNIP3-mediated CHIKV infection potential independently of autophagy) should be performed in other CHIKV permissive cell lines (such as HEK293T, HepG2, Hela)

Reviewer #2: The authors that examined the BNIP3 gene function in CHIKV replication. They are using siRNA approach for screening ATG-related gene and BNIP3 gene with viral infection. Then, they found that BNIP3 can increase CHIKV infection, but did not induce cell death and autophagy. Some comments as following:

1. If BNIP3 is not correlated to cell death or autophagy process, but why have some papers was demonstrated them. Here, the authors should provide more strongly evidences on this.

2. The authors may just said BNIP3 can regulate the CHIKV replication during virus replication cycle, but should have more viral protein expression profile.

3. In figure 3A, knock-downed the BNIP3 it seems to enhance the mitochondria broken-down in Mock+siBNIP3 and CHIKV+ siBNIP3 groups.

4. Autophagy process is a flow process should have some signaling pathways induction, such as mTOR/ULK1 or AKT/mTOR/ULK1 did not just said from gene expression level, which still have some protein modification such as protein phosphorylation.

5. If possible the authors should provide a cartoon to summary the results for more ease to check the point.

PLOS authors have the option to publish the peer review history of their article (what does this mean?). If published, this will include your full peer review and any attached files.

Reviewer #1: Yes: Domenico Mattoscio

Reviewer #2: No
---

## [Decision Letter · Decision Letter 1]

6 May 2023

Dear Dr. Smit:

Thank you very much for submitting your manuscript "BNIP3 regulates Chikungunya virus infection independently of autophagy and cell death" (PNTD-D-22-01058R1) for review by PLOS Neglected Tropical Diseases. 

As with all papers reviewed by the journal, your manuscript was reviewed by members of the editorial board and by several independent reviewers. Based on the reviews, we regret that we will not be pursuing this manuscript for publication at PLOS Neglected Tropical Diseases.

The reviews are attached below this email, and we hope you will find them helpful if you decide to revise the manuscript for submission elsewhere.

While we cannot consider your manuscript further for publication in PLOS Neglected Tropical Diseases, we would like to offer you the option to transfer your submission, with reviews, to PLOS ONE https://www.editorialmanager.com/PONE/

If you DO wish to transfer your submission, please click this link:

<DeepLinkData><DeepLinkTypeID>27</DeepLinkTypeID><peopleID>153881</peopleID><userSecurityID>b54c9d8a-ce9b-430d-b463-b1b84540f1ed</userSecurityID><documentID>29693</documentID><revision>1</revision><manuscriptNumber>PNTD-D-22-01058</manuscriptNumber><docSecurityID>c63b68f2-848c-4f6b-bcb8-5632419bbbba</docSecurityID></DeepLinkData>

If you do NOT wish to transfer your submission, please click this link to decline:

<DeepLinkData><DeepLinkTypeID>28</DeepLinkTypeID><peopleID>153881</peopleID><userSecurityID>b54c9d8a-ce9b-430d-b463-b1b84540f1ed</userSecurityID><documentID>29693</documentID><revision>1</revision><manuscriptNumber>PNTD-D-22-01058</manuscriptNumber><docSecurityID>c63b68f2-848c-4f6b-bcb8-5632419bbbba</docSecurityID></DeepLinkData>

Please note, all PLOS journals are editorially independent and vary in submission requirements.

Should you choose to transfer, your manuscript files, along with the reviewers' comments and their identities will be transferred automatically, and you will receive a confirmation email within 24 hours. Once transferred, your submission will be returned to you so you can check over your record before completing the submission. You may be asked to provide additional information, such as a response to the reviewers' comments. If you have any questions, please contact the editorial office of PLOS ONE https://www.editorialmanager.com/PONE/

We are sorry that the news is not more positive on this occasion, and we hope you will consider PLOS Neglected Tropical Diseases for future submissions. Thank you for your support of PLOS and of open-access publishing.

Sincerely,

Gregory Gromowski

Academic Editor

Abdallah Samy

Section Editor

Shaden Kamhawi

co-Editor-in-Chief

orcid.org/0000-0003-4304-636X

Paul Brindley

co-Editor-in-Chief

Reviewer's Responses to Questions

**Key Review Criteria Required for Acceptance?**

**Methods**

-Are the objectives of the study clearly articulated with a clear testable hypothesis stated?

-Is the study design appropriate to address the stated objectives?

-Is the population clearly described and appropriate for the hypothesis being tested?

-Is the sample size sufficient to ensure adequate power to address the hypothesis being tested?

-Were correct statistical analysis used to support conclusions?

-Are there concerns about ethical or regulatory requirements being met?

Reviewer #3: (No Response)

**Results**

-Does the analysis presented match the analysis plan?

-Are the results clearly and completely presented?

-Are the figures (Tables, Images) of sufficient quality for clarity?

Reviewer #3: (No Response)

**Conclusions**

-Are the conclusions supported by the data presented?

-Are the limitations of analysis clearly described?

-Do the authors discuss how these data can be helpful to advance our understanding of the topic under study?

-Is public health relevance addressed?

Reviewer #3: (No Response)

**Editorial and Data Presentation Modifications?**

Reviewer #3: (No Response)

**Summary and General Comments**

Reviewer #3: This is a revised version of the manuscript: “ BNIP3 regulated chikungunya virus infection independently of autophagy and cell death”, by Echavarria-Consuegra, et al., describes an autophagy independent role for BNIP3 as an inhibitor of chikungunya virus (CHIKV) replication. Overall, the manuscript is well written and the data generally supports the authors’ conclusions regarding BNIP3’s antiviral role. In general, the authors have also addressed the concerns raised in the prior reviews. However, there are a few points that still require attention.

Major points:

1) The study relies heavily on derived data, including viral titer results in Figures 5F and 6B. While the fold differences over control may be significant, it is difficult to tell how biologically meaningful these differences are without access to the actual titer data. Therefore, this non-normalized data should be provided either in the figures or as a supplemental table. 

2) Similar to point 1, the data on the effects of BNIP3 in MRC5 vs Hela or Huh7 cells should include the raw, non-derived infectivity data. If viral infection is significantly higher in control Hela or Huh7 cells compared to MRC5 cells, this could mask differences between scramble and BNIP3 cells, and the reader needs to be able to assess this.

3) Does CHIKV infection affect BNIP3 expression?

4) Since some of the data involves multiple comparisons (e.g. Fig. 2D, 3B), it should be clarified whether the statistical analysis took this into account, and it not, appropriate statistical tests performed.

Minor Point

1) The name chikungunya should not be capitalized in the title, since it is not a place name.

PLOS authors have the option to publish the peer review history of their article (what does this mean?). If published, this will include your full peer review and any attached files.

Reviewer #3: No

---

## [Decision Letter · Decision Letter 2]

5 Oct 2023

Dear Dr. Smit,

We are pleased to inform you that your manuscript 'Mitochondrial protein BNIP3 regulates Chikungunya virus replication in the early stages of infection' has been provisionally accepted for publication in PLOS Neglected Tropical Diseases.

Best regards,

Gregory Gromowski

Academic Editor

Abdallah Samy

Section Editor

---

## [Editor Report · Acceptance letter]

17 Nov 2023

Dear Dr. Smit,

We are delighted to inform you that your manuscript, " Mitochondrial protein BNIP3 regulates Chikungunya virus replication in the early stages of infection ," has been formally accepted for publication in PLOS Neglected Tropical Diseases.

Best regards,

Shaden Kamhawi

co-Editor-in-Chief

Paul Brindley

co-Editor-in-Chief
